# Fabrication of Inorganic Coatings Incorporated with Functionalized Graphene Oxide Nanosheets for Improving Fire Retardancy of Wooden Substrates

**DOI:** 10.3390/polym14245542

**Published:** 2022-12-18

**Authors:** Tsung-Pin Tasi, Chien-Te Hsieh, Hsi-Chi Yang, Heng-Yu Huang, Min-Wei Wu, Yasser Ashraf Gandomi

**Affiliations:** 1Division of Research and Development, Cite Engineering Consultants Ltd., Miaoli 350011, Taiwan; 2Department of Chemical Engineering and Materials Science, Yuan Ze University, Taoyuan 32003, Taiwan; 3College of Harbour and Environmental Engineering, Jimei University, Xiamen 361021, China; 4Department of Chemical Engineering, Massachusetts Institute of Technology, Cambridge, MA 02142, USA

**Keywords:** fire retardancy, wooden substrate, graphene oxide sheets, construction coating, surface functionalization

## Abstract

Flame-retardant chemicals are frequently used within consumer products and can even be employed as a treatment on the surface of different types of materials (e.g., wood, steel, and textiles) to prevent fire or limit the rapid spread of flames. Functionalized graphene oxide (FGO) nanosheets are a promising construction coating nanomaterial that can be blended with sodium metasilicate and gypsum to reduce the flammability of construction buildings. In this work, we designed and fabricated novel and halogen-free FGO sheets using the modified Hummers method; and subsequently functionalized them by pentaerythritol through a chemical impregnation process before dispersing them within the construction coating. Scanning electron microscopic images confirm that the FGO-filled coating was uniformly dispersed on the surface of wooden substrates. We identified that the FGO content is a critical factor affecting the fire retardancy. Thermogravimetric analysis of the FGO coating revealed that higher char residue can be obtained at 700 °C. Based on the differential scanning calorimetry, the exothermic peak contained a temperature delay in the presence of FGO sheets, primarily due to the formation of a thermal barrier. Such a significant improvement in the flame retardancy confirms that the FGO nanosheets are superior nanomaterials to be employed as a flame-retardant construction coating nanomaterial for improving thermal management within buildings.

## 1. Introduction

Fire is extremely destructive in nature and can cause serious damage to our lives and the community around us. Therefore, significant attention must be paid for improving fire safety regulations to further reduce fire hazards associated with combustible materials such as wood and textiles [1,2]. In recent years, numerous efforts have been dedicated to developing fire-resistant technologies/materials. Thus, flame-retardant chemicals have been employed in a variety of consumer products and adopted as a treatment on various types of surfaces (e.g., textiles, plastics, resins) to prevent or limit the spread of fire [3,4,5,6,7,8]. In essence, fuel, heat, oxygen, and free radical reactions are the four necessary components for the combustion of any material [9,10]. Flame retardants are generally used as an additive to polymers, fibers, and coating materials to increase their thermal stability and reduce flammability. Flame retardants are thus activated by ignition sources and prevent the growth of fire via eliminating/limiting any one of these key components [11].

Graphene sheets are two-dimensional (2D) nanocrystals with a single layer or several atomic layers consisting of a sp^2^-bonded carbon structure [12,13]. Several prior applications of graphene nanomaterials have been successfully demonstrated for transparent conducting films [14], catalyst supports [15,16], flow batteries [17], lithium-ion batteries [18], and electrochemical capacitors [19]. Because of ultra-high thermal stability, specific surface adsorption, as well as strong adhesion capabilities, graphite-like carbons (e.g., flaky graphite and functional graphene) can effectively alleviate heat and mass transfer [20,21,22,23,24,25,26,27,28,29,30]. The layered lamellar structure of graphene forms a barrier layer [31], limiting oxygen access along with reducing the heat transfer rate at the interface [32]. The graphene oxide (GO) nanosheet, one of the carbon allotropes, is frequently exfoliated from the natural graphite powders via chemical-wet exfoliation methods. The as-prepared GO sheets are usually composed of a large amount of oxygen functionalities, such as carboxyl (C–OOH), carbonyl (C=O), and hydroxyl (C–OH) groups [33]. It has already been reported that carbonyl and carboxyl groups tend to bond to the edge sites of the graphene sheet, while hydroxyl and epoxy groups are mostly attached to the basal planes [34]. The unique 2D carbon structure can be extensively functionalized by polymers or other inorganic compounds, forming a homogeneous fire retardant.

In this study, we report a novel, environmentally friendly, halogen-free, and low-toxicity functionalized GO (FGO)-filled flame retardant on wooden substrates. To reduce flammability, functionalized GO (FGO) nanosheet-filled construction coatings (including sodium metasilicate + gypsum) were deposited on wooden substrates. An efficient and relatively simple method was employed to functionalize the GO sheets through a chemical-wet impregnation method in the presence of pentaerythritol. The FGO content was shown to be a critical factor controlling the fire retardancy of the construction coatings on the wooden surfaces. The direct heating test confirmed significantly enhanced anti-flammability characteristics of the treated wooden substrates. During the combustion process, it was observed that the surface dehydrates, while releasing multiple gases (i.e., CO_2_, H_2_O), ultimately forms a char-like material that serves to protect the entire structure against the spread of fire.

## 2. Experimental Section

### 2.1. Synthesis of FGO Sheets

The GO sheets were exfoliated from natural graphite (NG) powders made by the modified Hummers method [35]. First, 5 g of NG powder was mixed with 120 mL of a strong oxidizing agent (1 M KMnO_4_ and concentrated H_2_SO_4_) in an ice bath. The resulting slurry was stirred continuously for 2 h, and the solution mixture was gradually heated until it reached 98 °C and then was maintained at this temperature for 15 min. The graphite slurry was then neutralized and went through several cycles of filtration and washing steps. The GO powders were then prepared via drying the slurry at 105 °C in an oven overnight. To prepare the FGO samples, the as-prepared GO sheets were functionalized by the pentaerythritol (chemical formula: C_5_H_12_O_4_, molecular weight: 136.15 g/mol, boiling point: 276 °C) through a chemical-wet impregnation. The chemical impregnation was performed in an ultrasonic bath at ambient temperature for 1 h to finalize the surface modification at the liquid phase.

### 2.2. Fireproof Performance of FGO-Filled Coatings

To examine the fire-retardant performance, the fireproof coating (including sodium metasilicate + gypsum) mixed with different amounts of FGO sheets was coated over a wooden plate. Herein, the wooden plates (i.e., three plywood) were carefully cut into a rectangular shape (4 × 5 cm^2^) with an average thickness of ~8 mm. Five different configurations, including 1, 3, 5, 7, and 9 wt% (i.e., the weight ratio of FGO sheets to the construction coatings) were implemented for preparing the samples. To enhance uniformity, the FGO-containing coatings were blended (for 10 min) using a three-dimensional mixer that included the zirconia balls. The as-prepared slurries were then pasted onto the wooden substrates with a doctor blade and dried at 40 °C in an oven overnight. The thickness of the fireproof coatings was controlled at ~1.5 mm (i.e., averaged over five different readings at various locations, with a deviation of ~0.1 mm). Herein, the FGO sheets had a lower apparent density (i.e., tap density: <1.0 g/cm^3^) compared to the other components present in the fireproof coating. This reveals that higher FGO content does not strongly alter the surface density of the nanocoatings. The flame retardancy of the construction coatings on wooden plates was evaluated using a high-performance flamethrower. The distance between the coating and the top of the flame was set at 5 cm, where the surface temperature on the wooden plates was kept at 1100 °C. During fireproof testing, three thermocouples (*K* type) were employed to measure the surface temperatures of the plates at various locations. The test procedure adopted in this work was identical to the “BS 476: Part 7” standard for examining the anti-flammability of the CaCO_3_ plates [36].

### 2.3. Materials’ Characterization

The morphology and structure of the FGO samples were characterized by field-emission scanning electron microscope (FE-SEM, JEOL JSM-5600) and high-resolution transmission electron microscope (HR-TEM, JEOL, JEM-2100). A thermos-gravimetric analyzer (TGA, Perkin Elmer TA7) and differential scanning calorimetry (DSC, TA Instrument Q20) were adopted to explore the thermal stability as well as the calorimetric change of FGO-filled coatings. A DSC measurement was carried out while heating the sample to 600 °C with a heating rate of 5 °C/min under an air atmosphere. The TGA analysis was also implemented in air (flowrate: 20 mL/min) with a heating rate of 5 °C/min, ramping from 50 °C to 700 °C.

## 3. Results and Discussion

Figure 1a,b shows the FE-SEM photographs of the resulting GO sheets with various magnification rates. According to Figure 1, the GO powder forms a dense structure with curled graphene-stacking sheets, generating a fluffy agglomeration. This suggests that the modified Hummers method can produce two-dimensional GO nanosheets from the chemical exfoliation of a NG precursor [37]. Based on the resulting GO structure, the FGO sheets still maintained a similar morphology even after the surface functionalization of the pentaerythritol, as depicted in Figure 1c,d. According to Figure 1, the FGO sheets contain numerous nanovoids and nanocavities. HR-TEM analysis was performed to further characterize the microstructures, as shown in the insets of Figure 1a,c. Herein, we observe that the pristine GO sample is composed of curved monatomic or several-layered sheets, confirming a layer-by-layer exfoliation process of the graphene sheets from the NG powders. Upon surface functionalization, the FGO sheets still maintain their transparent silky shape (~several square micrometers).

For consistency, throughout the entire paper, the FGO-filled construction coatings are labeled as G1, G3, G5, G7, and G9, according to 1, 3, 5, 7, and 9 wt.% FGO content within the fireproof layers, respectively. To explore the effect of surface functionalization, the fireproof coating filled with pristine GO sheets (1 and 9 wt.%) was also prepared for the comparison, as shown in Figure 2a,b, respectively. Herein, the fireproof coating had a roughened surface due to the poor uniformity of GO sheets, mainly caused by the GO agglomeration. Several cracks and GO agglomerates could be viewed on the surface structure, mainly due to low surface coverage of the phenolic groups on the pristine GO sheets that does not severely contribute to the hydrogen bonding between the sodium metasilicate and the pristine sheets. The agglomeration on the composite’s fracture surface can be attributed to the poor dispersion and weak interfacial interactions of the sodium metasilicate within the matrix. In contrast, the fireproof coatings with the addition of 1 and 9 wt.% FGO sheets (i.e., G1 and G9 samples) displayed a smoother surface without any apparent roughness (please see Figure 2c,d). This observation reveals that the incorporation of the FGO sheets via the surface functionalization of pentaerythritol in sodium metasilicate + gypsum mixture (as shown in Figure 3) leads to a homogeneous dispersion.

As for the FGO-filled composite coatings, the strong hydrogen bond interaction among FGO, gypsum (i.e., CaSO_4_·2H_2_O), and sodium metalsilicate (i.e., polymeric metasilicate anions [–SiO^2−^_3_–]_n_) tends to create homogeneously dispersed graphene sheets with strong interfacial interactions on the surface structure. Such an improved dispersion and interfacial interactions enhances the mechanical stability (i.e., anti-scratch property). Further increasing the FGO loading to 9 wt.% results in an even smoother layer with no obvious agglomerates, confirming the formation of a well-developed graphene network. These graphene nanostructures are capable of increasing the viscosity of a fireproof coating (suppressing dripping during combustion), along with the char yield (protective barrier layer), while restraining the flammability [38,39].

TGA analysis was also used to investigate the thermal degradation as well as the char formation [40,41]. Herein, the fireproof coating filled with different loadings of FGO sheets were subjected to a controlled temperature ramp from 50 to 700 °C at a heating rate of 5 °C/min. The TGA experiment was carried out in air, as shown in Figure 4a. The TGA curves for all the samples consist of three different thermal degradation steps: (i) 100–200 °C (dehydration), (ii) 200–350 °C (ignition and decomposition), and (iii) 550–700 °C (pyrolysis and degradation) [41]. Among all the samples, we observed that the major weight loss took place during Stage (i) while liberating water vapor from the mixture of gypsum + sodium metalsilicate (the weight loss at Stage (i) for all samples: 10–11 wt.%). The weight reduction during Stage (iii) can be attributed to the fact that amorphous carbon has evolved and has a maximal gasification rate at around 550 °C, whereas the maximum gasification rate of carbonaceous materials (graphite-like) occurs at ca. 650 °C [42,43]. It is worth noting that the decomposition delays with increased FGO content. It is generally recognized that the effectiveness of flame retardants can be evaluated via analyzing the volatiles, and the resulting residue with increased temperature [44]. In comparison, the residual weight after the thermal oxidation at 700 °C demonstrated the following order: G9 (63.8 wt.%) > G5 (59.8 wt.%) > G1 (57.5 wt.%). Accordingly, the integration of FGO sheets improves the anti-flammability of construction coatings. This improved residual yield as well as the reduced mass loss rate can be attributed to the addition of FGO sheets that facilitates the formation of an insulating layer within the composite structure. Increasing the FGO content, highly stable FGO sheets generate a so-called “tortuous path”, which further inhibits the release of volatile products and the mass/heat exchange during the thermal degradation process [45,46]. Based on the TGA analysis, the phenolic resin on the wooden fireproof coatings exhibits thermal degradation at 295–300 °C (weight loss: ~20 wt.%), as well as the two-stage weight loss (i.e., 350–370 and 475–485 °C with a weight loss: ~10 wt.%) under the nitrogen atmosphere. This weight loss is attributed to the phenolic resin curing (i.e., diverging between the resin formulations) and the thermal degradation [47]. The residual weight reaches ~48.3 wt.% after 700 °C, where the thermal resistance is lower than that of FGO-filled fireproof coating (i.e., high inorganic content, e.g., SiOx).

To further explore this trend, DSC analysis was performed to examine the heat flux as a function of temperature under a constant air flow, as depicted in Figure 4b. There is an obvious exothermic peak for all the samples; however, this exothermic peak shows a temperature lag with increased FGO content. The exothermic peaks take place at approximately 125 (G1), 142 (G5), and 175 °C (G9). Therefore, the thermal decomposition of FGO-filled fireproof coating due to the trailing effect significantly alters the ignition temperature [41]. Therefore, it can be deduced that the FGO sheets can serve as an effective fire-retardant additive for tailing away the decomposition reactions.

Figure 5 shows two sets of digital photographs for the fireproof coatings filled with 1 and 9 wt.% FGO sheets on the wooden substrates, where the as-prepared wooden substrates were heated to 1100 °C for 10 s. According to Figure 5, both samples contain obvious traces of burning on their corresponding surface structure. Herein, the area fractions for the pyrogenation (i.e., color: slight grey) and carbonization (i.e., color: black) can be considered as a crucial indicator to evaluate the flame retardancy. When heating up to 1100 °C for 10 s, the area fractions of pyrogenation and carbonization on the G1 sample were found to be much larger (compared to the G9 sample). This reveals that the addition of FGO sheets significantly alleviates the flame spreading and burning, decreasing both the pyrogenation and carbonization areas. To extend the applicability of this approach, the FGO-filled paste was also coated onto the stainless-steel foils to characterize the fireproof performance (i.e., at 1100 °C for 40, 50, and 60 s). Similarly, both the pyrogenation and carbonization areas reduced with increased FGO sheets, as shown in Figure 6. This observation also demonstrates the importance of FGO content on the fireproof performance. In addition, the FGO-filled coatings displayed excellent stability (without substantial weight loss and good adhesion to the substrates) and durability (no peeling from the substrates upon water washing) even after storing them in air for 6 months. This functional coating also exhibited superior thermal resistance and flame prevention (~1100 °C).

Figure 7 shows the carbonization fraction as a decreasing function of FGO content on wooden substrate after the flammability test was conducted at 1100 °C for 10 s. This improved fire retardancy can be ascribed to the fact that the chemical decomposition rate is retarded by the FGO filling within the composite coating. This trend also suggests that the thermal-oxidative stability of the char residue can be markedly improved by the introduction of FGO sheets, enhancing the fire resistance of the composite film [8,39,48]. It is generally recognized that graphene enhances the heat transfer in the composite coating and can also potentially decrease the thermal degradation temperature [49]. Indeed, the heat release is greatly reduced by the surface loading of FGO sheets if dispersed uniformly (see Figure 3). Since FGO sheets were well-dispersed in the fireproof coating, the resulting FGO sheets significantly enhanced the fire retardancy under the forced flaming condition.

Figure 8 illustrates variation of the carbonization fraction with the ignition time for both FGO-filled fireproof coatings on a wooden plate and the stainless-steel foil. As shown in Figure 8, both carbonization fractions were obviously alleviated with an increased FGO content. In other words, the dispersive FGO sheets led to high-flame-retardant efficiency. According to Figure 5 and Figure 6, when the FGO loading was increased to 9 wt.%, the graphene network combined with the sodium metasilicate + gypsum residues to form a close-grained char layer with reduced holes and/or smaller cracks. The resulting composite film served as an effective physical barrier to reduce the mass/heat exchange between the condensed and gas phases and prevented the escape of volatiles, enhancing the flame retardancy [39]. As compared to the FGO sheets with lower loading, the embedded FGO network in the resulting matrix imparted a higher melt viscosity while maintaining the shape of FGO-filled fireproof coating (i.e., without melt spreading during the flame propagation) [39]. This interaction reveals that the contact area between the fuel (matrix) and fire for the as-prepared composite coatings is insignificant and the fire transfer is effectively inhibited with the help of a FGO-filled coating. Indeed, the FGO framework serves as a protective and insulating barrier limiting heat and oxygen diffusion while protecting the underlying wooden substrate [48]. Given its simplicity, ease of fabrication, scalability, and low cost, we believe that the fireproof coating developed in this work has great potential for large-scale applications. Furthermore, our approach can be easily adopted to prepare composite thin films on various types of substrates for fabricating novel flame-retardant coatings.

## 4. Conclusions

This work demonstrated an effective approach to functionalize GO sheets for improving the dispersion of FGO in the sodium metasilicate + gypsum matrix through a chemical impregnation method. The resulting construction coatings on wooden plates and on the stainless-steel foils displayed superior fireproof characteristics. Here, the influence of FGO content on the fire retardancy of the construction coating was systematically investigated. FE-SEM images showed that the FGO-filled coating was uniformly dispersed on the surface of wooden substrates. TGA analysis of the FGO coating revealed that a high mass of char residue can be obtained at 700 °C, indicating enhanced thermal insulating performance. DSC analysis revealed that the appearance of the exothermic peak delayed with an increase in the FGO sheets, mainly due to the formation of a thermal barrier reducing the heat transfer. Accordingly, the FGO nanosheets can serve as an efficient coating nanomaterial for fire protection of construction materials. Such a significant improvement in flame retardancy suggests that FGO nanoparticles are excellent additives to flame-retardant construction coatings, thanks to their high efficiency, pollutant reduction, non-toxicity, low cost, and environmental friendliness.

## Figures and Tables

**Figure 1 polymers-14-05542-f001:**
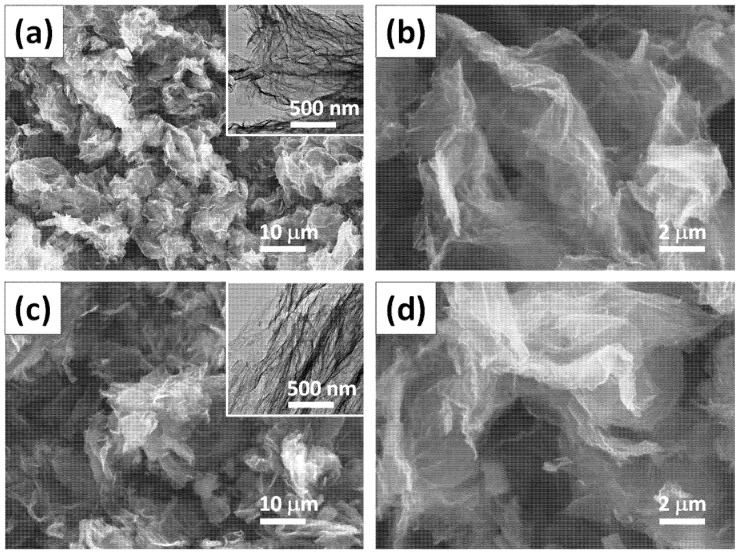
FE-SEM images of GO sheets with (**a**) low and (**b**) high magnifications. The inset of Figure 1a includes a HR-TEM micrograph of as-prepared GO sheets. FE-SEM images of FGO sheets with (**c**) low and (**b**) high magnifications. The inset of Figure 1c shows a HR-TEM micrograph of as-prepared FGO samples.

**Figure 2 polymers-14-05542-f002:**
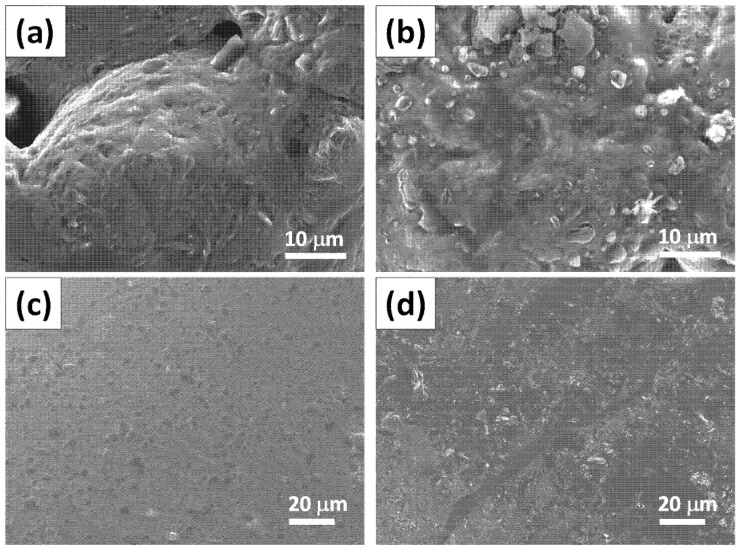
FE-SEM images of (**a**,**b**) pristine GO-filled coating and (**c**,**d**) FGO-filled coating with 1 and 9 wt.% FGO content, respectively.

**Figure 3 polymers-14-05542-f003:**
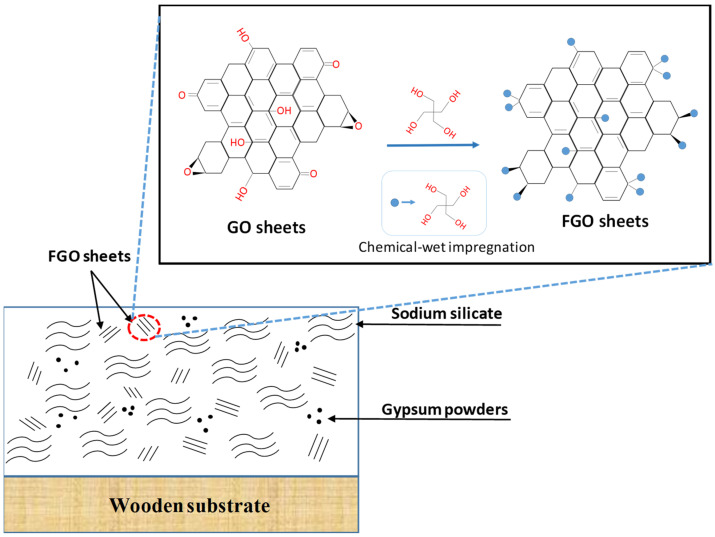
Schematic diagram of FGO-filled fireproof coating consisting of sodium silicate, gypsum powders, and FGO sheets, which were modified through the chemical-wet impregnation of pentaerythritol.

**Figure 4 polymers-14-05542-f004:**
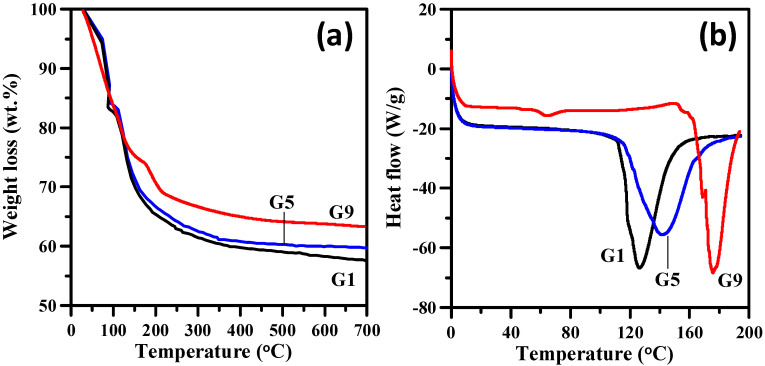
(**a**) TGA and (**b**) DSC curves of 1 wt.%, 5 wt.%, and 9 wt.% FGO-filled fireproof coating.

**Figure 5 polymers-14-05542-f005:**
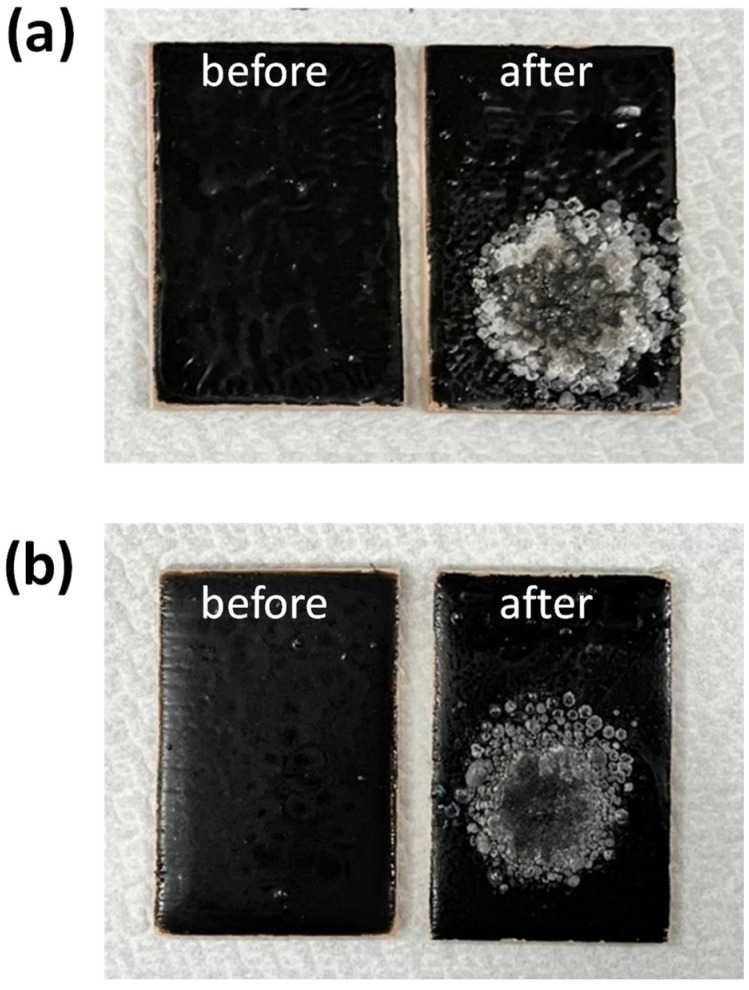
Photographs of (**a**) 1 wt.% FGO and (**b**) 9 wt.% FGO-filled fireproof coating on wooden plates, where “before” and “after” represents the plates before and after the high-temperature flammability test at 1100 °C for 10 s, respectively.

**Figure 6 polymers-14-05542-f006:**
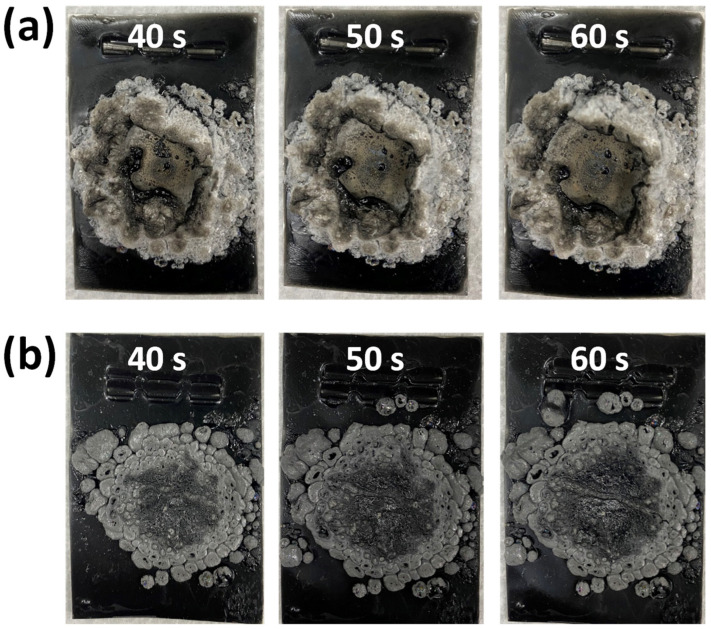
Photographs of (**a**) 1 wt.% FGO and (**b**) 9 wt.% FGO-filled fireproof coating on stainless steel foils, where the number represents the period of high-temperature flammability test at 1100 °C.

**Figure 7 polymers-14-05542-f007:**
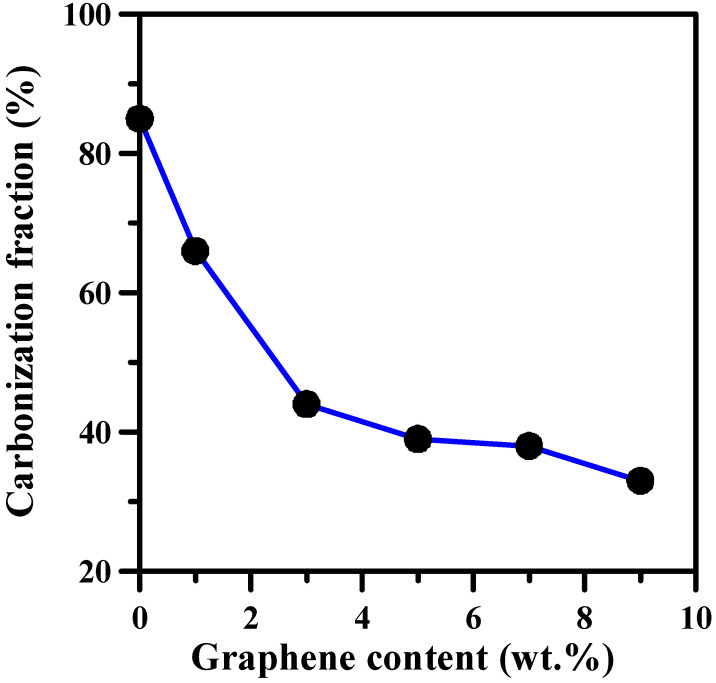
Carbonization fraction as a function of FGO content on wooden plates.

**Figure 8 polymers-14-05542-f008:**
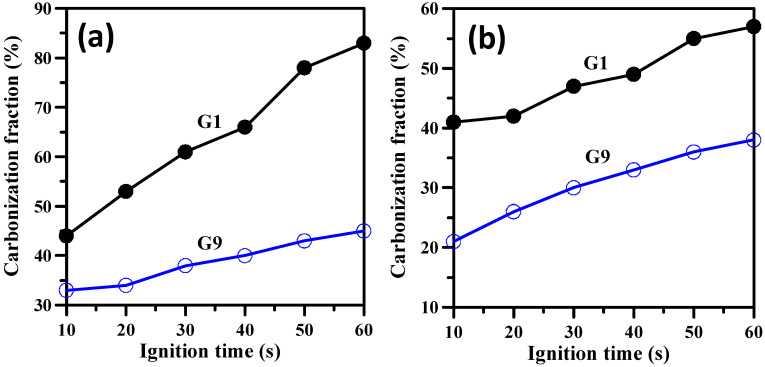
Variation of carbonization fraction with ignition time for both FGO-filled fireproof coatings on (**a**) the wooden plate and (**b**) stainless steel foil.

## Data Availability

Data is contained within the article.

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
