# Peer review of "Fabrication of Inorganic Coatings Incorporated with Functionalized Graphene Oxide Nanosheets for Improving Fire Retardancy of Wooden Substrates"

_polymers, 2022, doi:10.3390/polym14245542_

Round 1
Reviewer 1 Report
The presented work is well written, it presents the preparation of graphene in different proportions and its characterization by SEM, TGA and DSC. However, it does not present any variables that can be obtained from thermogravimetric analyses, weakening the work by not presenting these values ​​and, therefore, losing the opportunity to compare them with those of the literature.
It should strengthen the discussion of work, it is weak.
Both the TGA and the DSC are not explained in which atmosphere it has been carried out, if it is with air or Nitrogen, the data being different. Please comment on the differences and mention if it is in any of them. It is very important to mention the ramps made for the analyzes or experimental conditions.
Author Response
Response to Reviewer #1:
We would like to express our gratitude for the reviewer’s effort to improve the quality of this manuscript. What changes of the manuscript have been made are shown in the following responses to the specific comments. The corrections in the revision have been indicated in yellow background.
- Page 6-7. The reviewer’s concern is appropriate. We have collected literatures to compare their thermal resistance with the present work determined by TGA. The new references were cited at their appropriate locations of the revised manuscript. We have reflected them in the revision.
New references:
[37] M. Corrias, B. Caussat, A. Ayral, J. Durand, Y. Kihn, Ph. Kalck, Ph. Serp, Carbon nanotubes produced by fluidized bed catalytic CVD: approach of the process, Chem. Eng. Sci. 58 (2003) 4475-4482.
[38] C.-T. Hsieh, W.-M. Hung, W.-Y. Chen, J.-Y. Lin, Microwave-assisted polyol synthesis of Pt-Zn electrocatalysts on carbon nanotube electrodes for methanol oxidation, Inter. J. Hydrog. Energy 36 (2011) 2765-2772.
[42] P.L. de Hoyos-Martínez, H. Issaoui, R. Herrera, J. Labidi, F. Charrier-El Bouhtoury, Wood fireproofing coatings based on biobased phenolic resins, ACS Sustainable Chem. Eng. 9 (2021) 1729-1740.
- Page 4. The reviewer’s suggestion was adopted. To approach real conditions, both DSC and TGA analyses were carried out in air atmosphere (steady-state air flow). The experimental details concerning air flowrate (i.e., 20 mL/min) and ramping rate (5 °C/min) have been described in the revised manuscript. We have reflected it in the revised manuscript.
Reviewer 2 Report
Flame-retardant chemicals are frequently used within consumer products and can even be employed as treatments on the surface of different types of materials (e.g., wood, steel, and textiles) to prevent fire or limit the spread of fire. Among these materials, functionalized graphene oxide (FGO) nanosheet are promising construction coating nanomaterials that can be blended with sodium metasilicate and gypsum to reduce the flammability of construction buildings. In this work, authors proposed the synthesized functionalized FGO sheets prepared from the modified Hummers’ method and post-modification can be used as construction coating (on wood surface). It was shown that the FGO content is a critical factor affecting the fire retardancy. Thermogravimetric analysis of the FGO coating reveals that high char residue can be obtained at 700 °C. Differential scanning calorimetry indicates that the exothermic peak contains temperature delay in the presence of FGO sheets, primarily due to the formation of a thermal barrier. Such a significant improvement in the flame retardancy confirms that the FGO nanosheets are superior materials serving as the flame-retardant construction coating to improve the thermal management within buildings. I believe that this work can be accepted once the following raised concerns are fully addressed.
1. What is the thickness of FGO coating, compared to ~0.8 cm of wooden plates? How to control it smoothly? This information seems very significant.
2. Besides the difference of carbonization fraction (Figure 7), will the higher FGO contents (>9 wt%) leading to the significant enhancement of flame retardant? Basically, the higher FGO may result in the surface density of coatings.
3. In Figure 4a, it seems the FGO-filled fireproof coatings G9 contains less water inside (compared with G1 and G5, <100oC), making this comparison very weird. Please double check the comparison. “…The as-prepared slurries were then pasted on the wooden substrates with a doctor blade and dried at 40°C in an oven overnight…” Maybe the post-treatment temperature should be increased.
4. How about the stability of this FGO coatings? Can it be repeatedly applied to flame-prevention (~500 oC)?

Author Response
Response to Reviewer #2:
We would like to express our gratitude for the reviewer’s effort to improve the quality of this manuscript. What changes of the manuscript have been made are shown in the following responses to the specific comments. The corrections in the revision have been indicated in yellow background.
- Page 3. We adopted the reviewer’s suggestion to provide the information in the revision. The thickness of the fireproof coatings was controlled within approximately 1 mm. The thickness was averaged by collecting the data of five different locations, where its deviation was ±1 mm. We have reflected it in the Experimental section of the revised manuscript.
- Page 3. The reviewer’s inspection is valuable. We believe that higher FGO content would impart a significant enhancement of flame retardancy, which has been demonstrated by TGA and DSC analyses. In fact, the FGO sheets has a lower apparent density (i.e., tap density: < 1.0 g/cm3) compared to the other components in the fireproof coating. This reveals that higher content of FGO sheets would not strongly alter the surface density of coatings. We have briefly described it in the revised manuscript.
- Page 7. The reviewer’s concern is appropriate. Herein we have to emphasize the fireproof coating based on the recipe exhibiting a low-temperature, energy-saving, and efficient preparation. This is why we employ the post-treatment temperature at 40 °C. We also observe that an obvious removal of moisture at < 100 °C, mainly originated from the water molecules adsorbed onto the fireproof coatings. However, the weight loss from the water adsorption for all samples ranged from 10 to 11 wt.%, indicating the content of FGO sheets on the amount of moisture adsorbed is insignificant.
- Page 9. The reviewer’s comments are precious. The FGO-filled coatings displayed the excellent stability (without weight loss and good adhesion to the substrates) and durability (no peeling from the substrates after water washing) after storing them in air for 6 months. The functional coating also exhibited superior thermal resistance and flame prevention (~1100 °C) repeatedly. One brief description was added into the revised manuscript.
Reviewer 3 Report
I have reviewed the manuscript entitled "Fabrication of Inorganic Coatings Incorporated with Functionalized Graphene Oxide Nanosheets for Improving Fire Retardancy of Wooden Substrates" carefully. The authors experimentally studied functionalized graphene oxide (FGO) filled construction coatings with different wt%. The flame retardancy were tested. And the materials were characterized by SEM. HR-TEM, TGA and DSC. Their studies showed that the FGO nanoparticles are excellent additives for fire protection of construction materials. I think article is very interesting and well written, and it can be published as it is.
Author Response
Response to Reviewer #3:
We would like to express our gratitude for the reviewer’s effort to improve the quality of this manuscript. What changes of the manuscript have been made are shown in the following responses to the specific comments. The corrections in the revision have been indicated in yellow background.
- Thanks for the reviewer’s encouragement. The future work regarding the functional coatings applied to anti-electromagnetic applications on different substrates is in progress.
Round 2
Reviewer 1 Report
All changes requested from the authors were made. Thank you very much for doing them, the work has changed as soon as the discussion of the work, these changes have greatly improved the work presented.
Author Response
Response to Reviewer #1:
We would like to express our gratitude for the reviewer’s effort to improve the quality of this manuscript. What changes of the manuscript have been made are shown in the following responses to the specific comments. The corrections in the revision have been indicated in yellow background.
- We adopted the reviewer’s suggestion. The language of the manuscript has been moderately revised by a native speaker. The typos and errors have been corrected, and we have reflected them in the revision.
